# Dynamics of the Microbiota and Its Relationship with Post-COVID-19 Syndrome

**DOI:** 10.3390/ijms241914822

**Published:** 2023-10-01

**Authors:** Nidia Carolina Moreno-Corona, Orestes López-Ortega, Cesar Augusto Pérez-Martínez, Macario Martínez-Castillo, Luis Adrián De Jesús-González, Guadalupe León-Reyes, Moisés León-Juárez

**Affiliations:** 1Laboratory of Human Lymphohematopoiesis, Imagine Institute, INSERM UMR 1163, Université de Paris, 75015 Paris, France; moreno.corona.n@gmail.com; 2Université Paris Cité, INSERM UMR-S1151, CNRS UMR-S8253, Institute Necker Enfants Malades, 75015 Paris, France; olopezortega@outlook.com; 3Departamento de Física, Universidad de Sonora, Hermosillo 83000, Mexico; cesar.perez@unison.mx; 4Sección de Estudios de Posgrado e Investigación, Escuela Superior de Medicina, Instituto Politécnico Nacional, Mexico City 11340, Mexico; mmartinezcas@ipn.mx; 5Unidad de Investigación Biomédica de Zacatecas, Instituto Mexicano del Seguro Social, Zacatecas 98000, Mexico; adrian_6101@hotmail.com; 6Laboratorio de Nutrigenética y Nutrigenómica, Instituto Nacional de Medicina Genómica (INMEGEN), México City 16610, Mexico; greyes@inmegen.gob.mx; 7Laboratorio de Virología Perinatal y Diseño Molecular de Antígenos y Biomarcadores, Departamento de Inmunobioquímica, Instituto Nacional de Perinatología, Mexico City 11000, Mexico

**Keywords:** long COVID-19, SARS-CoV-2, gut microbiota, virus infection, post-acute COVID-19 syndrome

## Abstract

Coronavirus disease (COVID-19) is an infection caused by severe acute respiratory syndrome coronavirus 2 (SARS-CoV-2), which can be asymptomatic or present with multiple organ dysfunction. Many infected individuals have chronic alterations associated with neuropsychiatric, endocrine, gastrointestinal, and musculoskeletal symptoms, even several months after disease onset, developing long-COVID or post-acute COVID-19 syndrome (PACS). Microbiota dysbiosis contributes to the onset and progression of many viral diseases, including COVID-19 and post-COVID-19 manifestations, which could serve as potential diagnostic and prognostic biomarkers. This review aimed to discuss the most recent findings on gut microbiota dysbiosis and its relationship with the sequelae of PACS. Elucidating these mechanisms could help develop personalized and non-invasive clinical strategies to identify individuals at a higher risk of experiencing severe disease progression or complications associated with PACS. Moreover, the review highlights the importance of targeting the gut microbiota composition to avoid dysbiosis and to develop possible prophylactic and therapeutic measures against COVID-19 and PACS in future studies.

## 1. Introduction

Coronavirus disease (COVID-19) is an infection caused by the severe acute respiratory syndrome coronavirus 2 (SARS-CoV-2); it can be asymptomatic or produce mild symptoms such as fever, cough, sore throat, malaise, fatigue, nausea, vomiting, and diarrhea, with critical manifestations including respiratory failure, septic shock, and/or multiple organ dysfunction [1,2]. In addition, many people may exhibit chronic alterations associated with neuropsychiatric, endocrine, gastrointestinal, and musculoskeletal symptoms, even several months after disease onset, developing long-COVID or post-acute COVID-19 syndrome (PACS) [3,4].

The pathogenesis of PACS is unclear and may be multifactorial, involving prolonged inflammation, immune-mediated vascular dysfunction, thromboembolism, and nervous system dysfunction [5]. In addition, gut microbiota imbalance or dysbiosis may be critical for the emergence of PACS [6]. The human microbiota can be described as a diverse microbial community which can reside symbiotically in various anatomical sites of the human body [7]. SARS-CoV-2 mainly disrupts respiratory and gastrointestinal microbiota homeostasis in infected individuals [8,9,10,11,12]. This dysbiosis has detrimental effects on patients with COVID-19; for instance, it promotes a reduction in the abundance of beneficial symbiotic bacteria and an increase in the abundance of opportunistic pathogens [10,12]. Specifically, gut dysbiosis is associated with the occurrence of complications in COVID-19, disease severity, and persistent symptoms several months after initial infection [6,13,14,15,16]. 

Considering that microbiota imbalances may persist after recovery from acute COVID-19 symptoms, this review aims to summarize the dynamic relationship between gut microbiota communities and the pathophysiology of PACS.

## 2. Definition of Prevalence and Symptoms of Long COVID-19

On average, COVID-19 symptoms last between 1 and 4 weeks. However, a subgroup of patients worldwide present with persistent symptoms for many months [17]. These patients are diagnosed with long COVID or PACS [17]. 

PACS was first observed in the spring of 2020 when patients with COVID-19 continued to exhibit symptoms several weeks after their acute infections and soon after the first cases evolved [18]. According to the World Health Organization (WHO), PACS is defined as a condition that occurs with a history of probable or confirmed SARS-CoV-2 infection, generally 3 months from the time of COVID-19 diagnosis, with symptoms that last for at least 2 months and cannot be explained by an alternative diagnosis [19]. 

Individuals with PACS report a wide range of symptoms, including fatigue, shortness of breath, pain, mental disorders, cognitive dysfunction, and olfactory and gustatory dysfunctions. Symptoms may also fluctuate or relapse over time [18,19,20]. 

According to symptom severity, the following three clinical phenotypes have been described within PACS: permanence (unchanged during the follow-up period), relapse/remission (fluctuating episodes with intervals of exacerbation and remission), and slow progressive improvement [21]. 

The exact cause of PACS remains unknown; however, several hypotheses have been proposed, including direct viral tissue damage [22]. SARS-CoV-2 can access various tissues in the body, since the virus receptor, angiotensin-converting enzyme 2 (ACE2), is expressed by different cells [23]. Elevated C-reactive protein levels have been persistently detected during SARS-CoV-2 infection in some patients with PACS. For these particular cases, the etiology of the symptoms could be related to a specific increase in the CD8^+^ T cell response [23].

In addition to uncertainty about the etiology of PACS symptons, the prevalence in these patients remains unclear. The estimated prevalence depends on the follow-up duration, ethnicity, and symptoms defined in each PACS report [24]. Recently, community-based studies have suggested a low prevalence of persistent symptoms, while the majority of the people hospitalized following COVID-19 do not fully recover (50–70%) [25]. Several studies and meta-analyses have reported a wide range of prevalence for post-COVID-19. For example, a 2021 study of patients with COVID-19 who were followed up for up to 9 months after the illness revealed that approximately 30% of the patients exhibited persistent symptoms [17]. Another study estimated the possible incidence of post-COVID-19 sequelae to be between 10% and 35% [18]. Han et al. [26] reported that the most common symptoms at 1 year post-infection were fatigue/weakness, dyspnea, arthromyalgia, depression, anxiety, memory loss, concentration difficulties, and insomnia. Another study performed in a Danish population followed up to 3 months after COVID-19 diagnosis found that 39% of the patients had persistent symptoms and 8% of the patients had severe persistent symptoms, with the most common symptoms being fatigue, loss of smell, and taste dysfunction [20]. A study conducted in Italy found that 40.5% of the infected patients reported at least one of these symptoms 12 months post-infection [27].

As mentioned, the clinical presentation of patients with persistent COVID-19 is heterogeneous. More than 200 symptoms affecting different organs and systems have been reported. The most common long-term symptoms are fatigue (52%), cardiorespiratory symptoms (30–42%), and neurological symptoms (40%), including dysautonomia [21]. 

Attempts have been made to categorize the symptoms. One report classified symptom severity in acute and post-acute COVID-19 as follows: no severe multisystemic persistence of dyspnea (the main symptom in this category), followed by severe to critical COVID-19 when patients presented with multi-organ failure, and acute respiratory distress syndrome [28].

PACS heterogeneity is still unclear and the criteria for treatment and classification have evolved with time and with information availability. One factor involved in the recovery or susceptibility to PACS is related to the gut microbiota.

## 3. General Concepts of the Gut Microbiome and Its Association with the Lungs and Gastrointestinal Tract

The microbiome comprises the community of symbiotic, commensal, and pathogenic microorganisms of the human body. These microbes are particularly abundant in the gut, skin, and oral cavity [29]. 

Interestingly, although the gastrointestinal and respiratory tracts are anatomically and functionally different, the epithelia of the gut and lungs develop from a common embryonic structure and its microbial colonization is similar in early-life and in changes throughout life [30].

The gut microbiota is composed of bacteria (such as Firmicutes, Bacteroidetes, Actinobacteria, Proteobacteria, Fusobacteria, and Verrucomicrobia), fungi (Candida, Saccharomyces, Malassezia, and Cladosporium), viruses, phages, and archaea [31]. The lung microbiota includes the following four phyla: Bacteroidetes, Firmicutes, Proteobacteria, and Actinobacteria. The most abundant genera in healthy lungs are Prevotella, Streptococcus, Veilonella, Neisseria, Haemophilus, and Fusobacterium [32,33].

The gut microbiota participates in several functions, such as the fermentation of food, vitamin synthesis, immune system maturation, and protection against pathogens; hence, it is fundamental for maintaining health [31,34]. The lung microbiota avoids chronic inflammation, favoring the establishment of an immune-tolerant environment, particularly through the development, activation, and recruitment of immune-regulatory cells, such as regulatory T cells, M2-macrophages, and tolerogenic dendritic cells [33]. 

In addition to individual functions, crosstalk exists between the gastrointestinal tract and pulmonary microbiota which constitutes a functional gut–lung axis, which is essential for maintaining homeostasis and stimulating the host immune system in both organs [35]. For instance, gut segmented filamentous bacteria can activate the Th17 response, which modulates memory B cells in the lungs; conversely, the CCL25/CCR9 pathway induces the recruitment of lung-derived CD4+T cells in the intestinal tract, thus altering its microbiota [36,37]. Dysbiosis of the gut microbiota is associated with lung disorders and respiratory infections such as asthma and allergic inflammation. and in some patients with chronic lung disorders, irritable bowel syndrome develops as a consequence of gut microbiota disturbance [38,39].

This axis allows the passage of gut microbiota metabolites, endotoxins, and cytokines into the bloodstream connecting the intestinal tissues with the lungs. The main mechanism of this bidirectional regulation is the participation of bacterial metabolites such as short-chain fatty acids (such as acetate, butyrate, and propionate) that can modulate immune response activation by the engagement of G-protein coupled receptors and the inhibition of histone deacetylase (HDAC) activity in different cell types [38].

## 4. Gut Microbiome Association with Clinical Manifestations in Viral Infections

Viral infections attack several tissues and organs; human pathogenic viruses include those of the upper respiratory tract and lungs (e.g., adenovirus, influenza, rhinoviruses, and severe PACS-SARS-CoV-2), colon (e.g., enteroviruses, rotavirus, and norovirus), liver (e.g., hepatitis B virus), spinal cord (e.g., poliovirus), vascular endothelial cervix (e.g., human papillomavirus), cells (e.g., Ebola), and white blood cells (e.g., human immunodeficiency virus (HIV) and human T-cell leukemia virus) [40]. 

The mucosa is the main route of cellular entry from the environment [41]. The three main lines of defense on mucosal surfaces against pathogenic viruses are the mucus layer, innate immune defenses, and adaptive immune defenses [40,42]. Microbiota in the gut mucosa play an essential role in the immune response against viral infections by modulating both innate and adaptive immunity [40,41] 

For viral entry into the cell, the virus particle should attach to the host cell; some examples of how the microbiota inhibit this step are as follows. For the vesicular stomatitis virus, some strains of *Lactobacillus* and *Bifidobacterium* cause steric hindrance between the receptor and the virus. In addition, *Lactobacillus* expresses CD4 receptors that are capable of binding to HIV, thereby reducing viral infections [40]. Microbiota lipopolysaccharide (LPS) also exhibits antiviral mechanisms; immune cells primed with LPS show better activation mediated by toll-like receptor 4 (TLR4); thus, the influenza virus demonstrates decreased efficiency [43].

Metalloproteases produced by the microbiota play an essential role as antivirals. A study characterizing the genome of different populations of the S24_7 bacterial family identified M6 metalloproteases and immunoglobin A-degrading peptidase in some populations [44]. Microbial metabolites regulate the host immune system, and short-chain fatty acids (SCFAs) are the most essential metabolites of the gut microbiota. These SCFAs, including acetic acid, propionic acid, and butyric acid, are produced in millimolar quantities by the fermentation of dietary fibers [45]. Generally, SCFAs restrict the growth and adhesion of pathogenic microorganisms, improving epithelial integrity, and further enhance systemic host immunity by reducing intestinal pH, thus increasing mucin production [40,46]. For example, butyrate is associated with the reactivation of latent Epstein–Barr virus, due to its ability to regulate HDAC [47]. In addition, acetate protects against respiratory syncytial virus (RSV)-induced disease by enhancing the response to type 1 interferon (IFN) and increasing IFN-stimulated gene expression through activation of the membrane receptor GPR43 [48]. 

A common mechanism by which the gut microbiota influences antiviral control is by modulating the IFN response. Basal IFN secretion is necessary to maintain the immune system with constant vigilance for a robust response to infections [49,50]. 

It has been demonstrated that nasal and pharyngeal microbiota mediate susceptibility to influenza A (H3N2) and B infections. Staphylococci induce IFN production, which generates an innate response, resulting in a protective function of the microbiota against viruses [41]. In addition, desaminotyrosine (DAT), which is produced by a clostridial obligate anaerobe, promotes the synthesis of IFN-stimulated genes in the lungs and pulmonary phagocytes by augmenting type I IFN signaling [50]. Similarly, acetic acid produced by gut microbiota can inhibit RSV by inducing IFN-β production in the lungs, which activates the type I IFN signaling pathway [51]. Acetate also protects against RSV by activating the membrane receptor GPR43, which increases gene expression in response to type 1 interferons in the lung epithelial cells [48]. 

Another study mentioned that microbiota induces an enteric IFN-λ response in intestinal epithelial cells at homeostasis in the intestines of specific pathogen-free mice. Mice lacking the *Ifnlr1* gene are susceptible to murine rotavirus infections [52]. Glycolipids from *Bacteroides fragilis* signal through TLR4 to induce of IFN-β expression by colon DCs [53]. SCFAs and DAT produced by *Bacteroidetes* and/or *Clostridium* can enhance influenza-specific CD8+ T-cell function and type I IFN signaling in macrophages, thereby enhancing protection against influenza infection [54] As mentioned, microbiota in the lung–gut axis plays an essential role in mediating viral infections; therefore, microbiota dysbiosis may play an important role in COVID-19 infection.

## 5. Gut Microbiota and Its Dynamics in Typical COVID-19

Although SARS-CoV-2 infection is a pulmonary disease, ACE2 expression, the lung–gut axis, and antibiotic treatment can alter the gut microbiome [55]. COVID-19 is known to reduce the gut microbiome diversity in adult [13,56,57,58,59,60,61] and pediatric patients [62]. The loss of diversity in the gut microbiome is associated with other diseases [63], indicating the importance of the microbiome in health.

Specifically, dysbiosis caused by COVID-19 is characterized by a decreased abundance of commensal bacteria, including L-isoleucine and SCFA-producing bacteria [6,10,13,15,59,60,64,65]. Several beneficial effects of SCFAs have been reported, including anti-inflammatory properties and preventing colonic atrophy [66,67]. These effects are attributed to bacteria such as *Roseburia inulinivorans* [57], *Bifidobacterium adolescentis*, and *Faecalibacterium prausnitzii* [60]. Notably, *F. prausnitzii* has been underrepresented among patients with COVID-19 [6,10,13,15,60,64]. *F. prausnitzii*, which is associated with an anti-inflammatory effect independent of SCFA production [68], is positively correlated with an increased number of blood neutrophils [64], and negatively correlated with the severity of COVID-19 [6,13,15,60,64]. Other *Faecalibacterium* species are associated with fewer bloodstream infections [59]. Li et al. also reported an increase in the abundance of other butyrate-producing bacteria (*Paraprevotella* sp., *Streptococcus thermophilus*, *Clostridium ramosum*, and *Bifidobacterium animalis*) to compensate for the loss of SCFAs; however, this effect was not observed in other studies [57,60]. In summary, SCFA-producing bacteria play an essential role in the microbiome disrupted by SARS-CoV-2 infection, affecting the overall health and increasing the severity of infection.

In contrast, bacteria that were overrepresented during COVID-19 infection included the phyla *Firmicutes*, *Bacteroidetes* [59], and *Proteobacteria* in adults [13,58] and children [62]. *Proteobaceria* abundance is correlated with infection severity and fatality [58]. Several studies of patients with COVID-19 have reported the prevalence of opportunistic pathogens such as some species of *Clostridium* (*Clostridium ramosum* and *Clostridium hathewayi*) [10,57], *Escherichia*, *Veillonella*, *Rothia*, and *Streptococcus* [56,65]. Some bacteria, such as *Streptococcus thermophilus*, *Bacteroides oleiciplenus*, *Fusobacterium ulcerans*, and *Prevotella bivia*, are found only in patients with COVID-19 [57]. Interestingly, *Prevotella bivia* infection is correlated with low monocyte levels [57] and can cause bacterial vaginosis [69]. Moreover, patients with COVID-19 exhibit high gut levels of interleukin (IL)-18, a proinflammatory factor [70] that correlates with the prevalence of *Peptostreptococcus*, *Fusobacterium,* and *Citrobacter* [65]. Zhang et al. reported a correlation between the severity of COVID-19 and urea-producing bacteria [60]. However, no species have been implicated in this research. Overall, dysbiosis in patients with COVID-19 is characterized by an increase in the abundance of opportunistic bacteria, which correlates with the severity of infection. 

Although antibiotic treatment was once used empirically in COVID-19 treatment, it is no longer recommended [71,72]. Patients treated with antibiotics present more depletion of beneficial bacteria than patients with COVID-19 who have not been administered antibiotics [10]. In 2022, Schult et al. reported that antibiotic treatment was too diverse to determine its effect on the microbiota [13]. In addition, in 2022, Vestad et al. found similar microbiota in antibiotic-treated and non-treated patients with COVID-19 [61]. Moreover, Yeoh et al. identified that the microbiota from COVID-19-recovered patients who were not treated with antibiotics was more similar to the healthy control microbiota than those of the recovered patients treated with antibiotics [15]. Thus, considering these conflicting results, the effect of antibiotics on the microbiota of patients with COVID-19 needs further investigation.

Other respiratory infections also induce changes in microbiota composition and diversity [62,65,73], but they differ from the changes observed in patients with COVID-19. The reported differences are summarized in Table 1. Patients with severe RSV infections exhibited a greater abundance of the genera *Odoribacter* and *Oribacterium*, while those with moderate infections exhibited *Clostridiales* and *Coriobacteriaceae*, but no changes in microbiome diversity [74]. Patients with seasonal flu also exhibited a greater abundance of *Oribacterium*, *Bulleidia*, and *Aggregatibacter* species [65]. Patients infected with H7N9 showed no changes in the overall abundance of SCFA-producing bacteria. However, a decrease in the abundance of *F. prausnitzii* was observed in antibiotic-treated patients, but they shared a loss of diversity, a decrease in commensal bacteria, and an increase in the opportunistic bacteria observed in patients with COVID-19 [75,76]. A comparison between patients infected with COVID-19 and those with other respiratory infections revealed similarities in dysbiosis between patients with H1N1 and those with COVID-19, representing a loss of diversity and SCFA-producing bacteria. However, patients with COVID-19 exhibited increased abundance of opportunistic bacteria, such as *Streptococcus* and *Blautia*, whereas those with H1N1 exhibited an increased abundance of *Prevotella* and *Ezakiella* [56]. Moreover, COVID-19 induced the most dramatic loss of diversity, compared with other inflammatory diseases [62,65]. Together, these observations indicated a specific response of the gut microbiota when infected with SARS-CoV-2 which was not observed in infection with other respiratory viruses. 

Gut microbiota can affect the expression of ACE2 in human cells [77,78], thereby altering the entry of SARS-CoV-2 into intestinal cells. Zuo et al. reported that four *Bacteroides* species (*Bacteroides dorei*, *Bacteroides thetaiotaomicron*, *Bacteroides massiliensis*, and *Bacteroides ovatus*) negatively correlated with the viral load in the gut [10]. These species are associated with low ACE2 expression in mice [79]. *F. prautnitzii* and *Bacteroides thetaiotaomicron* reduce ACE2 expression in Caco-2 cells [80], and both bacteria are negatively correlated with COVID-19 [6,10,13,15,60]. Altering ACE2 expression can, in turn, affect amino acid uptake and other immunological processes [81]; these changes can affect disease severity, although this remains to be investigated. Thus, the absence of certain types of bacteria can increase ACE2 expression by cells in the gut; however, its effect on the overall disease remains to be determined.

In summary, changes in the gut microbiota during COVID-19 imply profound and specific alterations caused by the virus. These changes include an intense loss of diversity, an increased abundance of opportunistic bacteria, and a decreased abundance of commensal bacteria. These commensal bacteria include SCFA-producing species, such as *F. prautnitzii*. Changes in the overall abundance of different species can affect ACE2 expression, which in turn can affect disease severity. All these changes are expected in other respiratory infections, but patients with COVID-19 seem to be the most affected by these characteristics.

## 6. Gut Microbiota and Long COVID-19—Friend or Foe?

To date, the symptoms of COVID-19 have been widely described. However, the clinical features of patients with post-acute COVID-19 have not been fully elucidated. The possible explanations for the development of this syndrome are still unclear. The most probable hypothesis of this syndrome involves viral infection-induced altered inflammatory responses that could lead to exacerbated inflammation and cellular damage [82]. Additionally, intestinal microbiota changes have emerged as another potential explanation for these symptoms [6,10]. Several studies highlighted the involvement of the gut microbiota during infection, and even after its resolution (Table 2). Patients with COVID-19 exhibited significant changes in their fecal microbiome, displaying an enrichment in opportunistic pathogens and decreased abundance of beneficial bacteria [6,10] Gut commensals, such as *F. prausnitzii*, *Eubacterium rectale,* and *Bifidobacteria*, which possess immunomodulatory effects, are less abundant in patients during the active phase of the disease and one month after the infection has resolved. For example, the relative abundances of members of the genera *Ruminococcu*s and *Bifidobacterium* changed significantly in patients with COVID-19 and controls, with means of 3% vs. 6.75% for *Ruminococcus* and 16% vs. 19% for *Bifidobacterium*
**[6]**. At six months, patients with PACS showed significantly lower prevalences of *Collinsella aerofaciens*, *F. prausnitzii*, and *Blautia obeum* and higher prevalences of *Ruminococcus gnavus* and *Bacteroides vulgatus* than those of the controls [6,10] In addition, populations of opportunistic pathogens, such as Streptococcus anginosus, Streptococcus vestibularis, Streptococccus gordonii, and Clostridium disporicum, are abundant in patients with PACS [6,73,74]. These changes in the intestinal bacterial population modify the immune response based on contact with commensal organisms rather than opportunistic organisms, leading to elevated inflammation, since the gastrointestinal tract is an essential site of immune interaction between the host and pathogen, and may affect the recovery process of patients infected with COVID-19 [83]. Recently, F. prausnitzii has been described as possessing immunomodulatory properties, such as decreasing the inflammatory response by inhibiting the NF-κB pathway and, subsequently, the synthesis and secretion of IL-8, as well as inducing IL-10, which is an anti-inflammatory cytokine contributing to host defense.

A strong association between bacteria and COVID-19 is indicated by the fact that several Firmicutes bacteria can regulate ACE2 expression in the murine intestine [84]—possibly modulating viral infection [6,64,85]. According to these data, any alteration in the Firmicutes/Bacteroides ratio has a strong impact on the regulation of the immune response. These modifications in bacterial ratios have been observed in patients with COVID-19 and may play an important role in disease severity (Figure 1). In patients with mild, moderate, and severe COVID-19 symptoms, these ratios are 0.68, 0.65, and 0.58, respectively, indicating a correlation between the reduction in this ratio and disease severity [6,64,85].

**Table 2 ijms-24-14822-t002:** Recent studies of association between gut microbiota and long COVID.

Population	Study Groups	Age	Main Findings	Reference
China (Hong Kong)	PACS COVID-19 diagnostic patients (n = 106) non-COVID-19 controls patients (n = 68)	48.3 (33–62)	Patients without PACS showed recovered gut microbiome profile at 6 months comparable to that of non-COVID-19 controls. Gut microbiome of patients with PACS were characterized by higher levels of *Ruminococcus gnavus* and *Bacteroides vulgatus* and lower levels of *Faecalibacterium prausnitzii*. Persistent respiratory symptoms were correlated with opportunistic gut pathogens, and neuropsychiatric symptoms and fatigue were correlated with nosocomial gut pathogens, including *Clostridium innocuum* and *Actinomyces naeslundii*. Butyrate-producing bacteria, including *Bifidobacterium pseudocatenulatum* and *F. prausnitzii*, showed the largest inverse correlations with PACS at 6 months.	[57]
Italy (Roma)	Post SARS-CoV-2 positive (n = 31) SARS-CoV-2 negative (n = 18).	66.7 ± 14.4 67.1 ± 17.5	Bacteroidetes’ relative abundance was higher (≈36.8%) in patients with SARS-CoV-2 and declined to 18.7% when SARS-CoV-2 infection resolved (Six months). Firmicutes were prevalent (≈75%) in controls and in samples collected after SARS-CoV-2 infection resolution. *Ruminococcaceae, Lachnospiraceae* and *Blautia* increased after SARS-CoV-2 infection resolution six months after. *Lachnospiraceae (Fusicantibacter* and *Roseburia*) and *Ruminococcaceae* were increased after SARS-CoV-2 infection resolution.	[85]
Norway Oslo	Patients with 3 months after hospitalization COVID-19 (PACS) (n = 83)	59 (50–71)	Three months after hospitalization for COVID-19, patients with respiratory dysfunction showed a lower microbiota diversity and an altered global gut microbiota composition than patients with normal respiratory function. These microbiota alterations included reduced abundance of *Erysipelotrichaceae UCG-003* and increased abundance of *Veillonella* and *Flavonifractor.*	[61]
China (Hong Kong)	Post-acute COVID-19 syndrome (n = 302) Healthy controls (n = 893)	54.9	*Klebsiella pneumoniae*, an opportunistic pathogen, was positively associated with PACS patients. *Roseburia intestinalis*, a probiotic, was negatively correlated with PACS patients. Subjects with PACS showed a significant increase in abundance of *Bacteroides vulgatus* and *Bacteroides xylanisolvens*, compared with healthy controls.	[86]
China (Hong Kong)	Patients non-COVID-19 (n = 66) Patients COVID-19 (n = 66) With PACS (n = 48) Without PACS (n = 18)	47.9 (28–64) 49.2 (33–63)	Post-acute COVID-19-syndrome patients exhibited increased prevalence of *Klebsiella* sp.	[86]
United State of America (New Jersey)	Post-acute COVID-19 syndrome, single patient.	Not specified	*ASV002A_Bacteroides* was positively correlated with all PACS COVID-19 symptoms and positively correlated with the severity of anxiety. The *ASV0AKS_Oscillibacter, ASV009F_Anaerofustis, ASV02YT_Blautia, ASV07LA_Blautia*, and *ASV0AM6_Eubacterium hallii* amplicon sequence variants were potential SCFA-producing bacteria, which were associated with the alleviation of PACS COVID-19 symptoms.	[87]
Russia (Moscow)	post-COVID-19 syndrome patients (n = 30)	62 (53–67)	Excessive bacterial growth (92%) of proinflammatory microorganisms (*Bacteroides fragilis* group, *Candida* spp., *S. aureus*, *Proteus* spp., *Enterococcus* spp., *Enterobacter* spp., and *Citrobacter* spp.) exceeded the reference values by 1.5–2 times, and low levels of *Bacteroides thetaiotimicron* and *Akkermansia muciniphila* compared with the reference values. High ratio of *Bacteoides fragilis* group*/Faecalibacterium prausnitzii* associated with inflammatory diseases in post-COVID-19 patients.	[88]
China (Hong Kong)	Overall: COVID-19 patients n = 133 Cluster 1: severe and post-acute COVID-19 (n = 63) Cluster 2 non-severe COVID-19 (n = 70)	42.2 (26–59)	The multi-biome composition of patients in Cluster 1 was characterized by a predominance of bacteria *Ruminococcus gnavus, Klebsiella quasipneumoniae, Klebsiella pneumoniae, Klebsiella variicola, Erysipelatoclostridium ramosum, Clostridium bolteae,* and *Clostridium innocuum*, and lower relative abundance of *Bifidobacterium adolescentis* and *Faecalibacterium prausnitzii*.	[89]
China (Shanghai)	Post severe/critical COVID-19 infection patients (n = 14) Post mild/moderate COVID-19 infection patients (n = 31) Healthy controls (n = 31)	60.5 (49.5–70.5) 51.0 (42.7–56.2)	The diversity of the gut microbiome was reduced in severe/critical COVID-19 cases compared to mild/moderate cases. The abundance of some gut microbes altered post-SARS-CoV-2 infection and related to disease severity, such as *Enterococcus faecium*, *Coprococcus comes*, *Roseburia intestinalis*, *Akkermansia muciniphila*, *Bacteroides cellulosilyticus*, and *Blautia obeum*.	[90]
China (Wuhan)	Asymptomatic group (n = 103) long COVID-19 symptomatic patients (n = 84)	59 (48–66) 57 (45–67)	Symptomatic recovered patients (long COVID) had gut microbiota dysbiosis, including significantly reduced bacterial diversities and lower relative abundance of SCFAs-producing salutary symbionts, such as *Eubacterium hallii* group*, Subdoligranulum, Ruminococcus, Dorea, Coprococcus,* and *Eubacterium ventriosum* group, compared with healthy controls. The relative abundance of *Eubacterium hallii* group*, Subdoligranulum,* and *Ruminococcus* showed decreasing tendencies in healthy controls, the asymptomatic group, and the symptomatic long COVID group.	[91]

This association between microbiota and immune-system regulation is known; several studies have correlated gut microbiome alterations with chronic diseases, including gastrointestinal, inflammatory, metabolic, neurological, and respiratory diseases [31]. These findings in patients with long-term COVID-19 suggest a possible correlation between microbiota and some PACS symptoms, which may act as predictors of the development of disease-specific symptoms. 

Current evidence on the mechanism by which SARS-CoV-2 infection promotes microbiome dysbiosis is limited. However, the immunological response is hypothesized to be exacerbated during an acute infection, and the persistence or latent state of the virus in tissues, such as the intestine, may be related to post-COVID pathogenic events and microbiome dysbiosis [65,92]. Several viruses regulate the expression of genes involved in pro-inflammatory and anti-apoptotic processes, resulting in chronic inflammation. Additionally, various viruses that can persist in different anatomical regions and in biological fluids after acute infection has been identified [93]. For example, the Zika virus persisted for several months in the cerebrospinal fluid and lymph nodes of infected animals after passing through the convalescent phase [94]. The possible presence of SARS-CoV-2 for a long time in the intestinal tract of patients with COVID-19 could be contributing to the degradation of the intestinal mucosa through unexplored mechanisms [95,96]. Thus, some viral proteins can induce apoptosis in infected cells, or viral factors secreted by infected cells may participate in damaging neighboring cells; therefore, these mechanisms could be evaluated in the future to further understand how this microbial ecological deregulation occurs in the intestine.

## 7. Applications and Prospects of Gut Microbiota for the Diagnosis and Treatment of COVID-19 and PACS

The human microbiota plays a crucial role in maintaining host health by participating in the maturation, regulation, and induction of immune responses. The COVID-19 pandemic has highlighted the urgent need for innovative strategies to diagnose, manage, and treat the disease, as it poses unprecedented challenges to global healthcare systems [97]. Consequently, human microbiota has become a promising pathway for potential diagnostic, prognostic, and therapeutic applications, particularly in the context of COVID-19 and its long-term sequelae, PACS [98,99].

The dysbiosis observed in the gut microbiota and respiratory tract of patients with COVID-19 highlights the compelling association between the microbiota and the host immune response to the virus [100]. As described above, these perturbations in microbial composition have been linked to the severity of COVID-19 and post-COVID-19 manifestations and have been postulated as potential diagnostic and prognostic biomarkers. Such biomarkers may prove critical for identifying individuals at higher risk of severe disease progression or complications [101,102].

Identifying the microbial biomarkers correlated with COVID-19 severity may help in the development of personalized treatment strategies. The use of both intestinal [99,102,103] and airway [104,105,106,107,108] microbiota profiling techniques, such as metagenomic sequencing and metatranscriptomic profiling, could offer a comprehensive perspective on the population composition and functional activity of the microbial community and facilitate the development of microbial biomarkers to identify COVID-19 severity and PACS [109,110].

Furthermore, fecal microbiota signatures can be used as noninvasive diagnostic markers for COVID-19 [111]. The potential of microbiota-based diagnostics also extends to patients with PACS, where specific bacterial taxa have been reported to correlate with symptom severity (as discussed above). Microbiota-based diagnostic techniques could provide non-invasive tools to diagnose and manage PACS [112,113,114].

Moreover, intestinal microbiota modulation has been explored as a therapeutic option for COVID-19 or PACS through interventions such as probiotics, prebiotics, and fecal microbiota transplantation (FMT), which have shown promising therapeutic results [3]. These approaches may help to reduce inflammation, improve immune dysregulation, and decrease the risk of secondary infections in patients with COVID-19 and PACS [115,116]. However, further research is needed to establish standardized methodologies for microbiota-based therapies and to elucidate the underlying mechanisms that govern the complex interplay between gut microbiota and host immune response during COVID-19 infection.

The use of FMT as a treatment for COVID-19 and post-COVID-19 has been found to restore alterations in the microbiota and the intestinal immune response—even restoring the airway microbiota (through the intestine-lung axis). However, healthy donors must be selected, based on specific criteria such as physical and laboratory examinations [116,117,118].

In addition, Xavier-Santos et al. [115], through a systematic literature review, demonstrated that the administration of probiotics and prebiotics improved the composition of intestinal microbiota, reduced inflammation, improved vaccination efficacy, and prevented and/or alleviated the symptoms of COVID -19 and prolonged COVID-19. The species most commonly used in probiotic and prebiotic clinical trials are *Bifidobacterium longum* and *Bifidobacterium animalis* subsp. *Lactis, Bifidobacterium bifidum, Lacticaseibacillus rhamnosus, Lactobacillus acidophilus, Lactiplantibacillus plantarum, Limosilactobacillus reuteri, Loigolactobacillus coryniformis, Ligilactobacillus salivarius, Enterococcus faecium*, and *Saccharomyces cerevisiae* (Figure 2) [115,119,120].

The relationship between the gut microbiome and COVID-19 is complex and multifaceted. Understanding how the metabolites derived from this microbiome influence the disease may reveal new strategies for its treatment and prevention. During COVID-19 infection, metabolites such as glucose, glucosamine, galactose, and catechol are present in patients who are at a high risk of pulmonary complications. These metabolites are suggested to play a role in disease progression [121].

Moreover, SCFAs, such as acetate, butyrate, and propionate, which are primarily produced by bacteria of the *Clostridium* genus, are essential for the body’s immune response to SARS-CoV-2. These SCFAs can reduce ACE2 receptor expression, decrease viral load, and enhance adaptive immunity against viruses. Additionally, they possess anti-inflammatory properties that can mitigate excessive immune responses. In murine models, vancomycin has been shown to reduce the abundance of certain commensal bacteria and increase those of *Clostridium*, with the aim of increasing SCFAs. However, further human studies are required to determine their efficacy against COVID-19 (Figure 2) [122].

Notably, the ACE2 enzyme is vital for regulating the gut microbiota and strengthening innate immunity; however, it is also used by SARS-CoV-2 as a receptor to enter cells. Decreased ACE2 expression can compromise the intestinal barrier, leading to a leaky gut, which can exacerbate the cytokine storm in patients with severe COVID-19. Therefore, it is crucial to understand the interactions among the gut microbiome, SCFAs, ACE2, and SARS-CoV-2. This understanding could lead to innovative therapeutic approaches, such as gut microbiota modulation and SCFA supplementation, for patients with PACS, potentially improving clinical outcomes [123]. 

In summary, the intricate relationship between the gut microbiome and COVID-19 reveals several opportunities. Detailed investigations have revealed that the human gut microbiota has significant potential for devising new diagnostic, prognostic, and therapeutic techniques. Understanding the nuanced interplay between the microbiota and the host immune response to SARS-CoV-2 is paramount. Together, advancements in microbiota profiling and the identification of microbial biomarkers linked to COVID-19 severity hold promise for personalized treatment strategies for patients with COVID-19 and PACS.

## 8. Conclusions

Although PACS is a heterogeneous clinical syndrome with unclear causes, the hypothesis that dysbiosis of the gut microbiota participates in its pathogenesis has been proposed, based on recent data. These data suggest that a dynamic relationship could be established between the pathophysiological mechanisms activated by acute COVID-19 and PACS and intestinal microbiota. The displacement of commensal microorganisms by opportunistic pathogens is associated with detrimental complications and increased disease severity in infected patients. In addition, these data show the importance, in future studies, of targeting the gut microbiota composition to avoid dysbiosis and to develop possible prophylactic and therapeutic measures against COVID-19 and PACS.

## Figures and Tables

**Figure 1 ijms-24-14822-f001:**
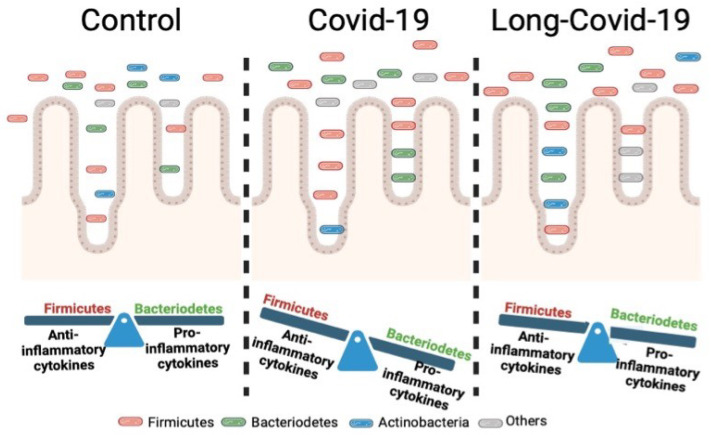
Changes in the microbiota as a function of the stage of COVID-19 infection. Alterations in the abundances of various types of bacteria, such as *Firmicutes* or *Bacteriodetes*, have been associated with the severity of infection and may have a patient-specific immunoimmunomodulatory function.

**Figure 2 ijms-24-14822-f002:**
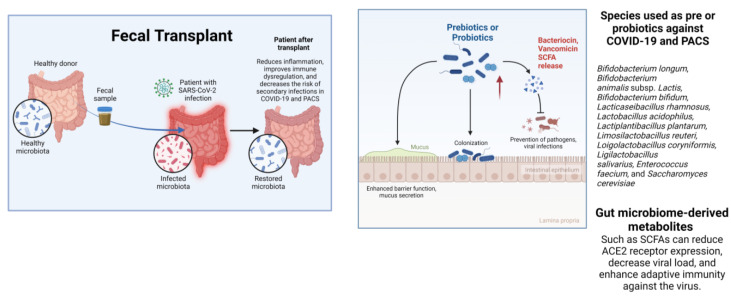
Therapeutic applications of microbiota for COVID-19 or PACS. One of the alternatives for treating secondary infections or PACS is fecal microbiota transplantation, which can restore the microbiota modified by the disease and restore the intestinal immune response. Additionally, probiotics and prebiotics can help improve the microbiota, reduce inflammation, and increase the effectiveness of vaccination.

**Table 1 ijms-24-14822-t001:** Differences in the dysbiosis produced by respiratory infections.

	Loss of Diversity	Increased Opportunistic Bacteria	Decreased Commensal Bacteria	Decreased SCFA Producing Bacteria	Decreased *F. prausnitzii*	Detected Specific Bacteria	References
COVID-19	+++	+	+	+	+	*Streptococcus thermophilus, Bacteroides oleiciplenus, Blautia, Fusobacterium ulcerans, and Prevotella bivia*	[6,10,13,15,57,59,60,64,65]
RSV	-	-	ND	-	-	*Odoribacter, Oribacterium*, *Clostridiales,* and *Coriobacteriaceae*	[74]
Seasonal flu	++	-	-	-	-	*Oribacterium*, *Bulleidia,* and *Aggregatibacter*	[60]
H1N1	+++	+	+	+		*Prevotella* and *Ezakiella*	[56,62]
H7N9	+	+	+	-	+ (Only in AB treated patients)	*Clostridium* sp., *Escherichia coli* and *Enterococcus faecium*	[75,76]

## Data Availability

Not applicable.

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
