# Peer review of "Dynamics of the Microbiota and Its Relationship with Post-COVID-19 Syndrome"

_ijms, 2023, doi:10.3390/ijms241914822_

Round 1

Reviewer 1 Report

The authors have undertaken a review of the current understanding regarding the impact of SARS-CoV-2 infection on the human microbiota. While the collection of studies included is deemed sufficient, there are substantial concerns regarding the quality of the review. In addition to requiring linguistic improvements, the overall structure of the narrative needs significant refinement to enhance readability.

Comments:

Line 49-50: Consider revising the sentence to something along the lines of, "The human microbiota refers to the diverse communities of microorganisms that coexist symbiotically at various sites within the human body."

Line 58: Suggest using "dynamic relationship" in place of the original term.

Line 64-70: To improve clarity, it is advisable to consistently use either "Long-COVID" or "PACS" throughout the text. Also, there is a discrepancy in the duration mentioned for PACS symptoms (2 vs. 3 months) within the same sentence.

Line 141-146: It would be helpful to provide evidence for the direct impact of SCFA (Short-Chain Fatty Acids) on the control of viral infections if the topic is mentioned. As it stands, this mention appears somewhat arbitrary.

Line 197: It's unconventional to mention an increase in Firmicutes and Bacteroidetes, as these are the two most abundant phyla. It would be more informative to specify which phyla decreased to accommodate the increase in Firmicutes and Bacteroidetes.

Line 230: Specify what "More similar to what?" refers to for clarity.

Table 1: Avoid making predictions about the impact of different viral infections on the microbiota based on only one or two research articles. Additionally, please correct the reference for article 48, as it does not pertain to seasonal flu.

Line 284-295: This paragraph exemplifies the difficulties in following the review. It jumps from discussing increased pathogens to Faecalibacterium, then skips to the Firmicutes/Bacteroides ratio without providing a coherent connection between these subjects.

Figure 1: Clarify the distinctions between the three epithelia depicted at the top of the figure for better understanding.

English language needs substantial improvement.

Author Response

24 September 2023

Dear Reviewers

I am pleased to resubmit for publication the revised version of “Dynamics of the microbiota and its relationship with post-COVID-19 syndrome”.  I appreciated the constructive criticism from the associated editor and reviewers. I have addressed each of their concerns as outlined below.      

Following the reviewer’s advice, I, along with my collaborators have been carefully revised and appropriate changes have been made in accordance with the reviewer’s suggestions. The responses to their comments are provided below:

We appreciate the recommendation and have contacted a service for correction of style and writing in English, through which the manuscript has been polished. In addition, a certificate of the company we hired will be attached.

Reviewer 1:

1.- Line 49-50: Consider revising the sentence to something along the lines of, "The human microbiota refers to the diverse communities of microorganisms that coexist symbiotically at various sites within the human body."

We have changed the phrase to: The human microbiota can be described as the diverse microorganism communities, that can reside in a symbiotic way in various anatomical sites of the human body.

 2.- Line 58: Suggest using "dynamic relationship" in place of the original term.

We change the term for: this review summarizes the dynamic relationship between gut microbiota communities

3.-Line 64-70: To improve clarity, it is advisable to consistently use either "Long-COVID" or "PACS" throughout the text. Also, there is a discrepancy in the duration mentioned for PACS symptoms (2 vs. 3 months) within the same sentence.

 We appreciate the comment, we have changed the term for PACS, and it has been highlighted among the text. We apologize for the misunderstanding of this phrase, the first 3 months is the time point when long covid is diagnosed and the 2 months refers to the duration of the long covid, we have rephrase as following; PACS is defined as a condition that occurs in with a history of probable or confirmed SARS-CoV-2 infection, normally 3 months from the time of appearance of COVID-19 with symptoms that last at least 2 months and cannot be explained by an alternative diagnosis.

4.-Line 141-146: It would be helpful to provide evidence for the direct impact of SCFA (Short-Chain Fatty Acids) on the control of viral infections if the topic is mentioned. As it stands, this mention appears somewhat arbitrary.

R: We agree that is helpful to provide evidence, for this reason we added two different examples of the control of  viral infection mediated by SCFA: In a general way SCFAs has been shown to reduce the growth and adhesion of pathogenic microorganisms, improve the integrity of the epithelium, and further enhance systemic host immunity by reducing intestinal pH, thus increasing mucin production [29,35]. For example the Butyrate has been associated  to the reactivation of latent Epstein-Barr Virus (EBV), due to its capacity to regulate HDAC[36]. Also, acetate was shown to protect against RSV-induced disease by enhancing the response to type 1 interferon and increasing interferon-stimulated gene expression through activation of the membrane receptor GPR43[37].

5.-Line 197: It's unconventional to mention an increase in Firmicutes and Bacteroidetes, as these are the two most abundant phyla. It would be more informative to specify which phyla decreased to accommodate the increase in Firmicutes and Bacteroidetes.

The line was changed to focus on the phylum Proteobacteria.

6.-Line 230: Specify what "More similar to what?" refers to for clarity.

The line was changed to: A comparison between COVID-19 patients and other respiratory infections reveals similarities between H1N1 patients and COVID-19 patients, meaning loss of diversity and loss of SCFA-producing bacteria; however, COVID-19 patients have more opportunistic bacteria, like Streptococcus and Blautia, whereas H1N1 patients have more Prevotella and Ezakiella [45].

7.-Table 1: Avoid making predictions about the impact of different viral infections on the microbiota based on only one or two research articles. Additionally, please correct the reference for article 48, as it does not pertain to seasonal flu.

R: The reference was corrected. The aim of the table was to inform of the reported differences, not make predictions. To make this clear, we added the text “The reported differences are summarized in Table 1” in line 223-224.

8.-Line 284-295: This paragraph exemplifies the difficulties in following the review. It jumps from discussing increased pathogens to Faecalibacterium, then skips to the Firmicutes/Bacteroides ratio without providing a coherent connection between these subjects.

 We appreciate the comment, we have edited the paragraph to make it easier to read and have the right connections. We added the next phrases: At six months, patients with PACS showed significantly lower levels of Collinsella aerofaciens, F. prausnitzii, and Blautia obeum and higher levels of Ruminococcus gnavus and Bacteroides vulgatus than controls [10,54]. In addition,populations of opportunistic pathogens, such as S. anginosus, S. vestibularis, S. gordonii, and Clostridium disporicum, have been described to increase in patients with PACS [6,73,74]. These changes in the intestinal bacterial population modify the immune response based on contact with commensal rather than opportunistic organisms, leading to elevated inflammation, as the gastrointestinal tract is an essential site of immune interaction between the host and the pathogen and may affect the recovery process of the COVID-19 infected patient [73]. Recently, F. prausnitzii has been described to have immunomodulatory properties, such as decreasing the inflammatory response by inhibiting the NF-κB pathway and subsequently the synthesis and secretion of IL-8, as well as inducing IL-10 - an anti-inflammatory cytokine - which contributes to host defence.

A strong connection between the bacteria-Covid relationship, is the fact that several Firmicutes bacteria can regulate the expression of ACE2 in the murine intestine - known to be a crucial molecule for virus entry[74] - possibly modulating viral infection [6,86,87]. According to these data, any alteration in the Firmicutes/Bacteroides ratio has a strong impact on the regulation of the immune response. These modifications in bacterial ratios have been observed in patients with COVID-19 and may play an important role in the severity of the disease (Figure 1). In patients with mild, moderate, and severe COVID-19 symptoms, these ratios are 0.68, 0.65, and 0.58, respectively, indicating a correlation between the reduction in this ratio and disease severity [6,86,87].

9.-Figure 1: Clarify the distinctions between the three epithelia depicted at the top of the figure for better understanding.

Thank you for your valuable comment, we have edited the figure to make it more informative and easier to analyze.

Reviwer 2 

The review article titled “dynamics of the microbiota and its relationship with post-COVID-19 syndrome” highlighted the recent findings on gut microbiota dysbiosis and its relationship with long-covid-19. Although, the review is interesting, addressing the following concerns would improve the article quality and significance.

 1.-Provide short information on general concepts of gut-microbiome and its link with GI tract and lungs.

We appreciate the observation; we corroborated that in the manuscript a new paragraph was added. This change can be noted in the final version of manuscript is highlight marked in yellow.

  1. Describe the potential mechanisms by which gut microbiota dysbiosis potentiate post-COVID 19 severity, specifically.

We appreciate the observation; we corroborated that in the manuscript a new paragraph was added. This change can be noted in the final version of manuscript is highlight marked in yellow.

  1. Gut microbiome-derived metabolites as therapeutic approach for post-COVID 19 patients, if any.

Thank you for your feedback, we have added a section on some explorations of microbiota-derived metabolites as a treatment for COVID-19. This change can be noted in the final version of manuscript is highlight marked in yellow.

  1. Figure 1 is not providing enough information, make it clear and more informative.

Thank you for your valuable comment, we have edited the figure to make it more informative and clearer.

Finally, we again thank you for your suggestions and insights, which have enriched the manuscript and produced a more balanced and better account of the review. We hope that the revised manuscript is now suitable for publication in the prestigious journal that you represent.

I look forward to your reply.

Sincerely,

Dr. León-Juárez Moisés

Departamento de Inmunobioquímica

Instituto Nacional de Perinatología ‘Isidro Espinosa de los Reyes’

Montes Urales #800, Col. Lomas de Virreyes

CP 11000, Ciudad de México, México.

01 55 5520 9900 ext-438

Reviewer 2 Report

The review article titled “dynamics of the microbiota and its relationship with post-COVID-19 syndrome” highlighted the recent findings on gut microbiota dysbiosis and its relationship with long-covid-19. Although, the review is interesting, addressing the following concerns would improve the article quality and significance.

1.     Provide short information on general concepts of gut-microbiome and its link with GI tract and lungs.

2.     Describe the potential mechanisms by which gut microbiota dysbiosis potentiate post-COVID 19 severity, specifically.

3.     Gut microbiome-derived metabolites as therapeutic approach for post-COVID 19 patients, if any.

4.     Figure 1 is not providing enough information, make it clear and more informative.

Quality of english is good

Author Response

(The authors gave the same response as above.)

Round 2

Reviewer 1 Report

Authors improved the first version of manuscript and is now suitable for publication.

Reviewer 2 Report

Authors answered all my queries.

English language looks good.